# Patent Blue V Dye Adsorption by Fresh and Calcined Zn/Al LDH: Effect of Process Parameters and Experimental Design Optimization

**Aicha Machrouhi [1], Nawal Taoufik [1], Alaâeddine Elhalil [2], Hanane Tounsadi [3], Zakia Rais [3] and Noureddine Barka [1,***

1   Multidisciplinary Research and Innovation Laboratory, Sultan Moulay Slimane University of Beni Mellal, FP Khouribga, 25000 Khouribga, Morocco; machrouhi.aicha90@gmail.com (A.M.); nawal.taoufik@gmail.com (N.T.)
2   Laboratory of Process and Environmental Engineering, Higher School of Technology, Hassan II University of Casablanca, 20000 Casablanca, Morocco; elhalil.alaaeddine@gmail.com
3   Laboratoire d'Ingénierie, d'Electrochimie, de Modélisation et d'Environnement, Faculté des Sciences Dhar El Mahraz, Université Sidi Mohamed Ben Abdellah, 23000 Fes, Morocco; hananetounsadi@gmail.com (H.T.); raiszakia@yahoo.fr (Z.R.)
*   Correspondence: barkanoureddine@yahoo.fr; Tel.: +212-661-66-66-22

**Abstract:** This work focuses on the adsorptive removal of patent blue V (PBV) dye from aqueous solution by Zn/Al layered double hydroxide in fresh (LDH) and calcined (CLDH) forms. The material was synthesized via coprecipitation and samples were characterized by XRD, FTIR and TGA-DTA. Dye retention was evaluated under different experimental conditions of contact time, pH, adsorbent dosage, temperature and initial dye concentration. Experimental results show that highest adsorption capacity occurred at acidic medium. Kinetics data were properly fitted with the pseudo-second-order model. Equilibrium data were best correlated to Langmuir model with maximum monolayer adsorption capacities of 185.40 and 344.37 mg/g, respectively, for LDH and CLDH. The process was endothermic and spontaneous in nature. Based on the preliminary study, full factorial experimental design ($2^4$) was used for the optimization of the effect of solution pH, adsorbent dose, initial dye concentration and the calcination. Thus, the optimal conditions to reach high equilibrium adsorption capacity were achieved at pH of 5, adsorbent dosage of 0.1 g/L, and initial dye concentration of 15 mg/L by CLDH.

**Keywords:** patent blue V; calcined LDH; sequestration; experimental design

## 1. Introduction

Water is an essential substance for daily life. Due to the expansion of industrial, agricultural and domestic activities, this substance is highly contaminated by industrial sludge, heavy metals, pesticides, organic dyes and other chemicals. This results in serious diseases affecting biomes and biota. According to the UNESCO World Water Assessment Program (WWAP), 100 million people, 1 million sea birds and 1 lakh marine mammals die each year due water pollution [1]. As ones of the most water polluting industries, textile and paper industries emit many organic dyes during the coloring process [2]. Even a tiny amount of these dyes can affect the quality of water. Among the dyes, patent blue V from the class of azo dyes is commonly used in cosmetic and food industry, which provides direct contact with human body. It is known to cause headache, asthma and allergic reactions [3,4]. However, various technologies are used in the treatment of wastewaters containing the excess level of dyes, including photodegradation [5], chemical coagulation [6], biodegradation [7], catalytic reduction [8] and electro-chemical treatment [9]. Among these processes, adsorption is proven to be one of the most attractive and effective techniques [10]. Therefore, several materials have attracted considerable attention as adsorbents in dye removal

applications. Layered double hydroxides (LDH) or anionic clays are known by their high retention capacity for anionic dyes [11] and are easily synthesized, less expensive, regenerable, large surface area, super anion exchange capacity and environmental friendliness. These compounds can be described by the general formula: $[M^{II}_{1-x}M^{III}_x(OH)_2]^{x+}.(A^{n-}_{x/n}).mH_2O$, where $M^{II}$ represents a divalent cation ($Mg^{2+}$, $Zn^{2+}$, $Ni^{2+}$, $Mn^{2+}$, $Fe^{2+}$ … ), $M^{III}$ represents a trivalent cation ($Al^{3+}$, $Cr^{3+}$, $Fe^{3+}$, $Co^{3+}$, $Mn^{3+}$ … ), $A^{n-}$ the compensating anion ($Cl^-$, $NO^-_3$, $ClO^-_4$, $CO^{2-}_3$ … ), n the charge of the anion, and m is the number of water molecules located in the interlayer region together with the anion. The coefficient, x, is the molar fraction, $[M^{III}/(M^{II} + M^{III})]$ [12].

The main purpose of this work was to evaluate the potential of fresh and calcined Zn/Al LDH as adsorbent for the removal of PBV dye from aqueous solution. Various factors such as contact time, solution pH, adsorbent dose, temperature and initial dye concentration were studied. This paper also investigated the combined effect of the most influencing parameters, which are solution pH, adsorbent dose, initial dye concentration and nature of adsorbent. Full factorial experimental design with two levels ($2^4$) was used to acquire the optimal conditions for high removal efficiency.

## 2. Materials and Methods

All used chemicals were of analytical grade and were used without further purification. Zinc nitrate ($Zn(NO_3)_2 \cdot 6H_2O$) ($\geq 99\%$), aluminum nitrate ($Al(NO_3)_3 \cdot 9H_2O$) ($\geq 98\%$), sodium carbonate ($Na_2CO_3$) (99.5–100.5%), sodium hydroxide (NaOH) ($\geq 99\%$), hydrochloric acid (HCl) (37%), sodium chloride NaCl (99.5%) and patent blue V ($C_{27}H_{31}N_2NaO_7S_2$) (100%) were obtained from Sigma-Aldrich (Germany). Bidistilled water was used as the solvent throughout this study. The characteristics and chemical structure of the dye are listed in Table 1.

**Table 1.** Molecular structure and physical characteristics of patent blue V.

| Name | Molecular Structure | $M_W$ (g/mol) | $\lambda_{max}$ (nm) |
|---|---|---|---|
| Patent blue V (Acid blue 3) |  | 582.66 | 637 |

LDH was synthesized by several authors previously, using the coprecipitation method at constant pH [12,13]. A mixture solution of $Zn(NO_3)_2 \cdot 6H_2O$ and $Al(NO_3)_3 \cdot 9H_2O$ with a total concentration of metal ions (Zn and Al) of 2 mol/L and $Na_2CO_3$ (1 mol/L) was added drop-wise in a backer containing 50 mL of bidistilled water. The pH of the mixture solution was adjusted and kept constant at $8.5 \pm 0.2$ during the synthesis by addition of suitable amounts of NaOH solution (2 mol/L). The formed gel was stirred vigorously for 4 h and then transferred into an autoclave and hydrothermally treated at 75 °C for 16 h. Finally, the precipitate was washed several times with deionized water until the solution was neutral and dried at 100 °C for 24 h. The resulting product (LDH) was ground into fine powder and stored in sample bottle for further use. Part of the resulting material was calcined at 500 °C in a tubular furnace for 6 h to obtain CLDH. Powder XRD patterns of the samples were recorded in 2θ range from 5 to 70° at room temperature on a D2 PHASER diffractometer, using CuKα radiations with 30 KV and 10 mA. FT-IR spectra were recorded in the range of 4000 to 400 $cm^{-1}$ using a Perkin Elmer (FTIR-2000) spectrophotometer. The sample was prepared by finely mixing 1 mg of adsorbent with 100 mg of KBr to prepare pellet.

Thermogravimetric and differential thermal analysis (TGA-DTA) curves were recorded on a SETARAM (SENSYSevo) instrument, in the temperature range from 30 to 700 °C with a heating rate of 10 °C/min under argon atmosphere. The pH of point of zero charge (pHpzc) was determined according to the method described by Noh and Schwarz [14]. The pH of NaCl aqueous solution (50 mL at 0.01 mol/L) was adjusted to successive initial values in the range of 2–12 by addition of $HNO_3$ and/or NaOH. Moreover, 0.05 g of each biosorbent was added in the solution and stirred for 6 h. The final pH was measured and plotted vs. the initial pH. The $pH_{pzc}$ was determined at the value for which $pH_{final} = pH_{initial}$.

Adsorption experiments were conducted at different conditions by adding different dosages (0.05–0.8 g/L) of LDH or CLDH to 100 mL of dye solution of predetermined concentrations (15–150 mg/L). The initial solution pH was varied in the range of 3–12, the contact time was in the range of 2–120 min, and temperature was in the range of 25–55 °C. After each adsorption experiment completed, the solid phase was separated from the liquid phase by centrifugation at 3000 rpm for 10 min. Then, the dye concentration was determined using a TOMOS V-1100 UV type Spectrophotometer.

The adsorption yield (%Removal) of the dye and the adsorption capacity (q (mg/g)) were evaluated by the following equations:

$$\%Removal = ((C_0 - C)/C_0) * 100 \tag{1}$$

$$q = (C_0 - C)/R \tag{2}$$

where $C_0$ and C are respectively initial and residual dye concentration a time t (mg/L), and R is the adsorbent dosage (g/L).

Based on the effect of each individual factor, the most influencing factors were used for the optimization of the process by experimental design. Table 2 shows the four factors used and their levels. The experiments were performed according to a full factorial design at two levels ($2^4$), with 16 experiments. The adsorption behavior was optimized by using a first-order polynomial model (Equation (3)):

$$Y = b_0 + b_1A + b_2B + b_3C + b_4D + b_{12}AB + b_{13}AC + b_{14}AD + b_{23}BC + b_{24}BD + Yb_{34}CD + b_{123}ABC + b_{124}ABD + b_{134}ACD + b_{234}BCD + b_{1234}ABCD \tag{3}$$

where, Y is the response of interest (adsorption capacity of PBV dye).

**Table 2.** Process factors and their levels.

| Factors | Levels | |
|---|---|---|
| | Low (−) | High (+) |
| A. Adsorbent dosage (mg/g) | 0.1 | 0.3 |
| B. Solution pH | 5 | 7 |
| C. Dye concentration (mg/L) | 15 | 30 |
| D. Nature of adsorbent | LDH | CLDH |

## 3. Results and Discussion

### 3.1. Characterization

#### 3.1.1. X-ray Diffraction (XRD) Study

The XRD patterns of the fresh and calcined Zn-Al-LDH are shown in Figure 1. The figure exhibits the characteristic reflections of the LDH structure for fresh sample with planes (003), (006), (012), (015), (018), (110) and (113). Remarkable changes were observed after calcination at 500 °C. The lamellar structure collapsed and new peaks corresponding to ZnO oxide started to appear indicated by the peaks at 2θ = 31.8°, 34.5°, 36.3°, 47.6°, 56.6°, 62.9°, 66.4°, 68° and 69.1°. These peaks correspond to the reflections from (100), (002), (101), (102), (110), (103), (200), (112) and (201) planes, respectively. This is also confirmed by the JCPDS data (Card No. 36-1451) [15].

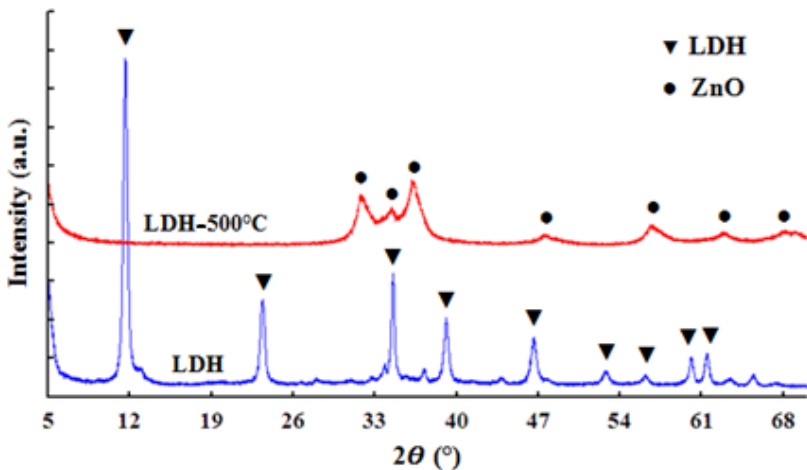

**Figure 1.** XRD patterns of LDH material before and after calcination at 500 °C.

3.1.2. Fourier Transform Infrared (FTIR) Analysis

Figure 2 shows the FTIR spectra of LDH before and after calcination. The spectrum of the fresh LDH shows a broad band between 3600 and 3200 $cm^{-1}$, which is attributed to the stretching vibration of the OH groups of physically adsorbed and interlamellar water molecules [16]. Another common band for the LDH materials is found at about 1600 $cm^{-1}$, attributed to the O-H bending vibrations of water molecules [16]. The band at 1364 $cm^{-1}$ is assigned to the stretching vibration of the $CO_3^{2-}$ groups in the LDH interlayer [16]. This band rapidly disappears after calcination due to the thermal decomposition of carbonate ions. Bands around 700–400 $cm^{-1}$ could be related to the lattice vibration modes such as the translation vibrations by M-O (590 and 670 $cm^{-1}$) and O-M-O (430 $cm^{-1}$) [17,18].

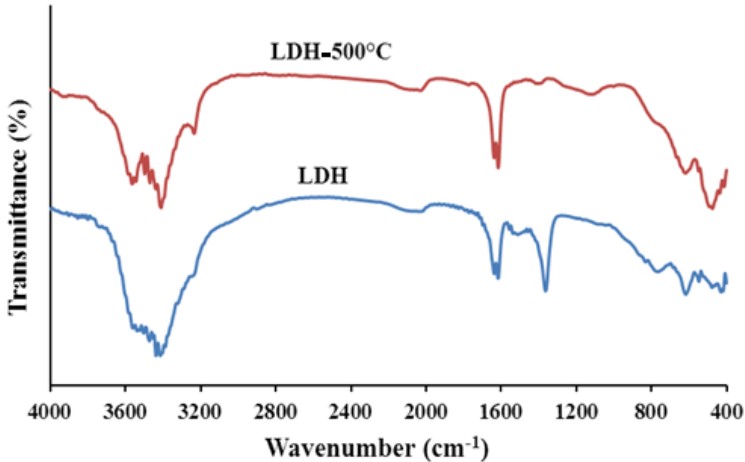

**Figure 2.** FTIR spectra of LDH material before and after calcination at 500 °C.

3.1.3. Thermal Analysis (TGA-DTA)

The thermal decomposition of the LDH was investigated by TGA-DTA analysis. According to the literature, the decomposition of LDH includes three main stages which are the loss of adsorbed water, the decomposition of $H_2O$, OH and finally $CO_3$. The TGA-DTA curves of LDH material obtained shown in Figure 3 are quite similar to those reported in previous research works [19]. The TGA-DTA curves of synthesized LDH show a first mass loss at ~100 °C, which can be accredited to the loss of adsorbed water. It was followed by a second more pronounced and sharp endothermic phenomenon around 160 to 240 °C. This mass loss was due to loss of hydration water from the interlayer region. A third step, extending up to 320 °C, is assigned to the overlapped mass losses due to the dehydroxylation of the layers and the decomposition of the carbonates.

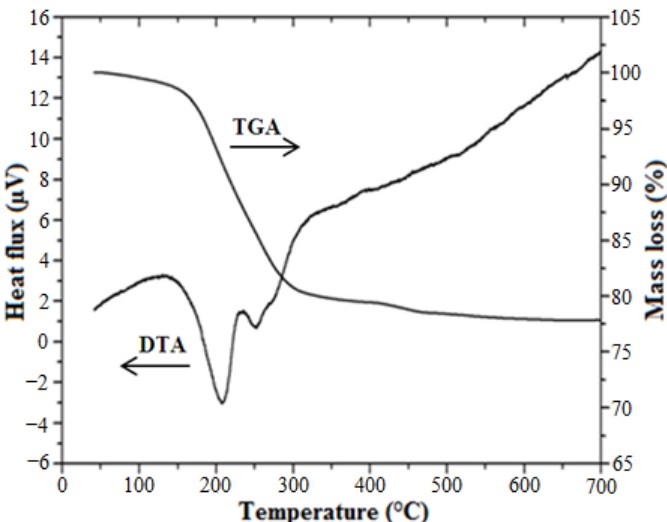

**Figure 3.** TGA/DTA curve of LDH material.

*3.2. Dye Removal from Aqueous Solution*

3.2.1. Effect of Solution pH

The effect of solution pH on the removal of PBV was investigated in a pH range varying from 3 to 12 as shown in Figure 4. It was observed that the highest adsorption was obtained in the pH range of 3 to 5 and then continually decreased with pH increase. The pH change would directly affect the negatively and positively charged site distribution profile on the surface of the adsorbents. The pH point of zero charge (pHpzc) of the adsorbents and the pKa value of dye molecule are important factors controlling this behavior. The pKa of PBV is 2.78, displaying that the dye molecules present as monovalent anions in solution in the studied pH range. The pHpzc of adsorbents were 7.47 and 8.1 for LDH and CLDH, respectively. Therefore, at pH > pHpzc the surface charge is negative, disfavoring the anions forms of dye molecules adsorption. At pH < pHpzc, the surface of adsorbents becomes more positively charged leading to an increase in the dye removal due to strengthening attractive forces among the positive charge of the surface of the adsorbents and negative charge of the dye molecules.

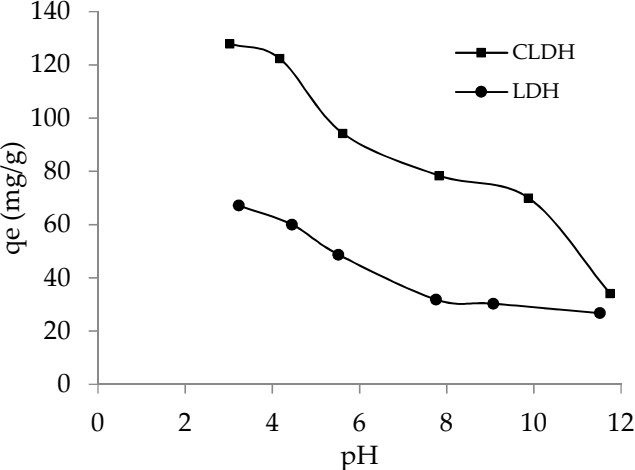

**Figure 4.** Effect of pH on the adsorption of PBV: $C_0$ = 15 mg/L, R = 0.1 g/L, agitation time = 2 h, T = 25 °C.

3.2.2. Effect of Adsorbent Dosage

Figure 5 represents the effect of adsorbent dosage on PBV removal by fresh and calcined LDH. The figure shows that the effectiveness of retention increases with the

increase of adsorbent dosage from 0.05 g/L to a value of 0.4 g/L, where it shows a plateau. This result is due to the fact that the increase in the adsorbents dosage increases the number of adsorption sites available for non-adsorbed dye molecules. The figure also shows that the removal efficiency does not reach 100% despite the continuous increase of adsorbent dosage behind 0.4 g/L. This suggests low to moderate interaction of the adsorbents by the dye molecules at low concentration in solution. From the figure, it can be also concluded that the affinity of PBV dye to CLDH is greater than that of fresh LDH.

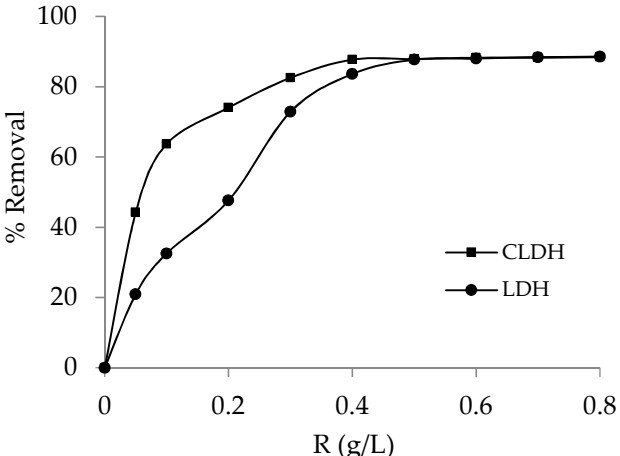

**Figure 5.** Effect of adsorbent dosage on the adsorption of PBV: $C_0 = 15$ mg/L, agitation time = 2 h, initial pH = 5.68 and T = 25 °C.

3.2.3. Adsorption Kinetics

Contact time is an important design parameter on which depends adsorption process as it provides information about the dynamic of the reaction in terms of order and of the rate constant. As shown in Figure 6, the removal capacity of PBV by CLDH and LDH both exhibited a rapid increase in the first 10 min of reaction time, then, gradually increased until reaching equilibrium at 30 min. Adsorption kinetics data were analyzed using pseudo-first-order model (Equation (4)) [20] and pseudo-second-order model (Equation (5)) [21].

$$q = q_e \left(1 - e^{-K_1 t}\right) \tag{4}$$

$$q = (K_2 \, q_e^2 \, t)/(1 + K_2 \, q_e \, t) \tag{5}$$

where $q_e$ and $q$ (both in mg/g) are, respectively, the amounts of dye adsorbed at equilibrium and at any time t (min), $k_1$ (1/min) is the rate constant of pseudo-first-order model and $k_2$ (g/mg min) is the rate constant of pseudo-second-order model.

Parameters of the pseudo-first-order and pseudo-second-order models were estimated with the aid of the nonlinear regression. The obtained data and the correlation coefficients, $r^2$, are given in Table 3. The Table indicates that the $r^2$ values are higher and closer to 1 for the pseudo-second-order model compared to those of the pseudo-first-order model. The calculated equilibrium values ($q_{cal}$) of the pseudo-second-order model are consistent with the experimental data. This result suggests that the adsorption of PBV onto LDH and CLDH could be better described by the pseudo-second-order model instead of pseudo-first-order kinetic model.

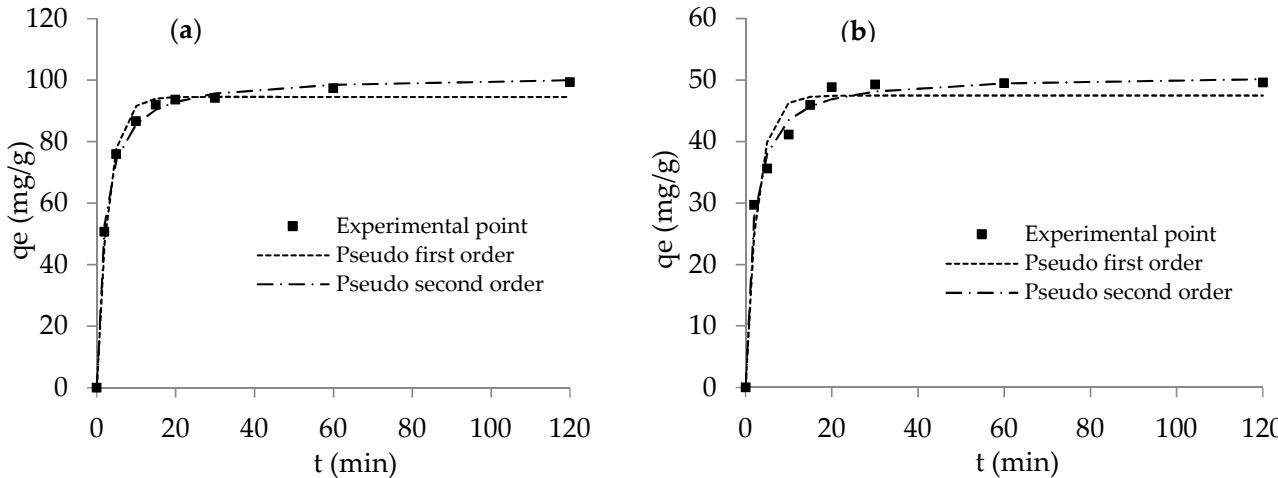

**Figure 6.** Adsorption kinetics of PBV by (**a**) CLDH and (**b**) LDH: $C_0 = 15$ mg/L, R = 0.1 g/L, initial pH = 5.52 and T = 25 °C.

**Table 3.** Kinetics models constants for PBV adsorption by CLDH and LDH.

| Adsorbent | $q_{exp}$ (mg/g) | Pseudo First-Order | | | Pseudo Second-Order | | |
|---|---|---|---|---|---|---|---|
| | | $q_{cal}$ (mg/g) | $K_1$ (1/min) | $r^2$ | $q_{cal}$ (mg/g) | $K_2$ (g/mg min) | $r^2$ |
| CLDH | 99.36 | 94.52 | 0.347 | 0.991 | 101.47 | 0.005 | 0.998 |
| LDH | 49.48 | 47.47 | 0.368 | 0.959 | 50.83 | 0.012 | 0.990 |

3.2.4. Adsorption Isotherms

Adsorption isotherm plays a crucial role in analyzing the adsorption capacity of materials and providing information about solution-surface interaction. The adsorption capacities of LDH and CLDH with increasing the initial dye concentration are shown in Figure 7. The figure indicates that the adsorbed amounts of PBV increased with the increase in equilibrium dye concentration. It's evident that high concentration in solution implicates high dye molecule fixed at the surface of the adsorbent. However, when the equilibrium concentration exceeded 60 mg/L, the adsorbents reached a saturated state. Obtained equilibrium data were analyzed using Langmuir (Equation (6)) [22] and Freundlich (Equation (7)) [23] isotherm models.

$$q_e = (q_m \, K_L \, C_e)/(1 + K_L \, C_e) \tag{6}$$

$$q_e = K_F \, C_e^{1/n} \tag{7}$$

where $q_m$ (mg/g) is the Langmuir maximum monolayer adsorption capacity, $K_L$ (L/mg) is the Langmuir equilibrium constant related to the adsorption affinity, $K_F$ ($mg^{1-1/n}g^{-1}L^{1/n}$) is the Freundlich constant related to the adsorption capacity and n is the heterogeneity factor related to the adsorption intensity.

The fitting parameters and correlation coefficient values ($r^2$) are all listed in Table 4. According to the $r^2$ values, the more suitable model for this system is the Langmuir model. This result indicates that the adsorption processes is more likely to be homogeneous and monolayer. The obtained $q_m$ values indicate that the adsorption capacity of CLDH (344.37 mg/g) is much higher than of LDH (185.40 mg/g). In addition, $K_L$ value obtained for CLDH is higher compared to that obtained for LDH, which confirm the high affinity of PBV dye to calcined LDH. This increase in the adsorption capacity of LDH with calcination could be attributed to the reconstruction phenomena. The calcination of LDH structure leads to the decomposition of the carbonate anion $CO_3^{2-}$ in the interlayer space and the formation of mixed oxides with memory effect. When these mixed oxides are

hydrated in solution, they reconstruct LDH structure by intercalation of dye molecules in its intermalleolar space [24].

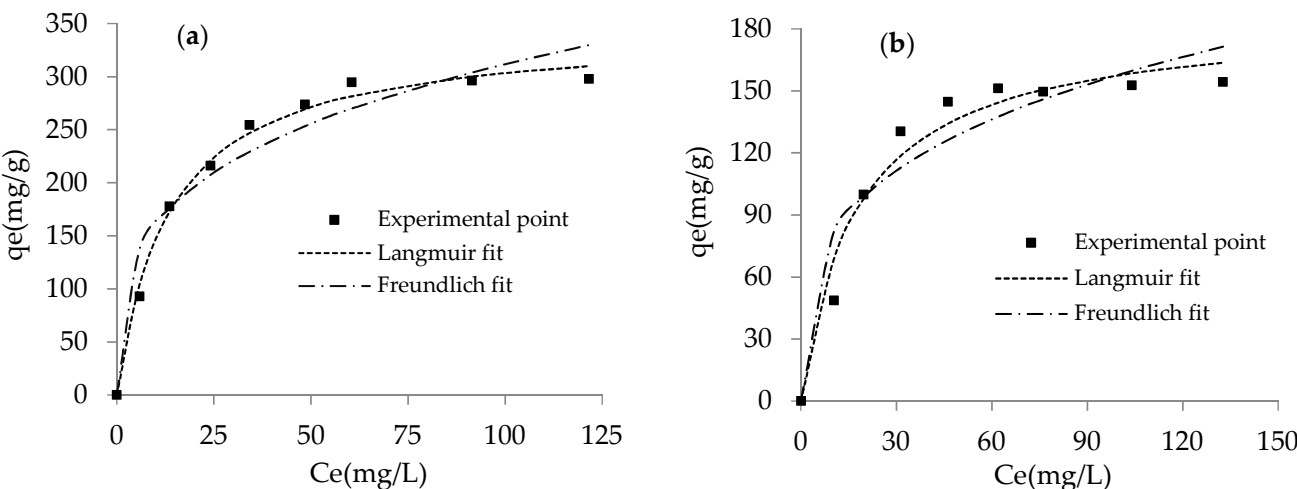

**Figure 7.** Experimental points and nonlinear fitted isotherm curves of PBV adsorption by (**a**) CLDH and (**b**) LDH: R = 0.1 g/L, agitation time = 2 h, initial pH = 5.62 and T = 25 °C.

**Table 4.** Isotherm models constants calculated for PBV adsorption by Fresh and calcined LDH.

| Adsorbent | Langmuir | | | Freundlich | | |
|---|---|---|---|---|---|---|
| | $q_m$ (mg/g) | $K_L$ (L/mg) | $r^2$ | $K_F$ (mg$^{1-1/n}$/g·L$^{-1/n}$) | n | $r^2$ |
| CLDH | 344.37 | 0.074 | 0.993 | 83.231 | 3.487 | 0.943 |
| LDH | 185.40 | 0.057 | 0.967 | 41.608 | 3.453 | 0.906 |

### 3.2.5. Effect of Temperature

The changes observed in the adsorption of PBV by varying solution temperature in the range of 25–55 °C are shown in Figure 8. From the figure, it can be seen that the adsorption yield was enhanced by rise in temperature. Thermodynamic parameters; $\Delta H°$, $\Delta G°$, and $\Delta S°$ were applied to assess the spontaneity and heat exchange during the adsorption process. They were calculated using the following equation:

$$LnK_D = -\Delta G°/RT = -\Delta H°/RT + \Delta S°/R \qquad (8)$$

where $K_D$ is the distribution constant of dye between solid phase and liquid phase ($q_e/C_e$), R is the universal gas constant (8.314 J/mol K), T is solution temperature in K, $\Delta G°$ is the Gibbs free energy, $\Delta S°$ is the entropy, and $\Delta H°$ is enthalpy. $\Delta S°$ and $\Delta H°$ were estimated from the slope and intercept of the plot of $lnK_D$ vs. 1/T yields.

The calculated parameters are illustrated in Table 5. The negative values of $\Delta G°$ and positive values of $\Delta S°$ imply that the removal process of PBV on both LDH and CLDH is feasible and spontaneous. Meanwhile, $\Delta H° > 0$ further demonstrates the endothermic nature of the process. Additionally, the values of $\Delta H°$ are lower 40 kJ/mol, suggesting physical adsorption process.

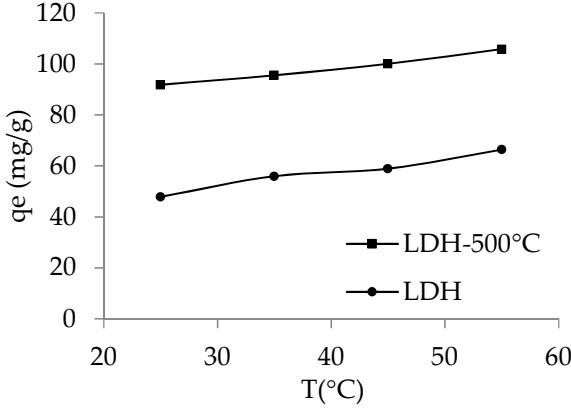

**Figure 8.** Effect of temperature on the adsorption of PBV by CLDH and LDH: $C_0$ = 15 mg/L, agitation time = 2 h, initial pH = 5.57 and R = 0.1 g/L.

**Table 5.** Thermodynamic parameters of the adsorption of PBV by CLDH and LDH.

| Adsorbent | T (K) | $K_D$ | $\Delta G°$ (kJ/mol) | $\Delta H°$ (kJ/mol) | $\Delta S°$ (J/K·mol) |
|---|---|---|---|---|---|
| CLDH | 298 | 16.48 | −6.94 | 11.67 | 62.41 |
| | 308 | 18.36 | −7.46 | | |
| | 318 | 21.06 | −8.09 | | |
| | 328 | 25.31 | −8.82 | | |
| LDH | 298 | 4.79 | −3.89 | 13.71 | 59.01 |
| | 308 | 6.10 | −4.63 | | |
| | 318 | 6.64 | −5.01 | | |
| | 328 | 8.18 | −5.73 | | |

### 3.3. Process Optimization

### 3.3.1. Experimental Results

The experimental results obtained at the designed conditions according to the full factorial experimental design are presented in Table 6. According to the table, the highest adsorption capacity of 92.2 mg/g was obtained by CLDH at initial dye concentration of 15 mg/L, pH of 5, and adsorbent dosage of 0.1 g/L. Statistical analysis was used to determine a well-fitted regression model. Values of the main effects of individual variables and their interaction effects obtained are presented in Table 7. From the table, it can be observed that the calcination of LDH has a positive effect on PBV adsorption. Meanwhile, the other remaining individual terms present a negative effect on the adsorption. The analysis of the interaction effects shows that the most significant interaction was between adsorbent dosage and initial dye concentration with a positive effect ($b_{13}$ = +4.93), followed by the interaction between adsorbent dosage and the nature of LDH ($b_{14}$ = −6.93).

**Table 6.** Factorial experimental design matrix in coded and real values and experimental results.

| Run | Coded Values | | | | Actual Values | | | | $q_e$ (mg/g) |
|---|---|---|---|---|---|---|---|---|---|
| | A | B | C | D | A | B | C | D | |
| 1 | −1 | −1 | −1 | −1 | 5 | 0.1 | 15 | LDH | 48.62 |
| 2 | −1 | 1 | −1 | −1 | 5 | 0.3 | 15 | LDH | 32.36 |
| 3 | 1 | −1 | −1 | −1 | 7 | 0.1 | 15 | LDH | 26.67 |
| 4 | 1 | 1 | −1 | −1 | 7 | 0.3 | 15 | LDH | 20.45 |
| 5 | −1 | −1 | 1 | −1 | 5 | 0.1 | 30 | LDH | 15.90 |
| 6 | −1 | 1 | 1 | −1 | 5 | 0.3 | 30 | LDH | 22.00 |
| 7 | 1 | −1 | 1 | −1 | 7 | 0.1 | 30 | LDH | 11.81 |
| 8 | 1 | 1 | 1 | −1 | 7 | 0.3 | 30 | LDH | 14.20 |
| 9 | −1 | −1 | −1 | 1 | 5 | 0.1 | 15 | CLDH | 92.24 |
| 10 | −1 | 1 | −1 | 1 | 5 | 0.3 | 15 | CLDH | 40.47 |

**Table 6.** *Cont.*

| Run | Coded Values | | | | Actual Values | | | | $q_e$ (mg/g) |
|---|---|---|---|---|---|---|---|---|---|
| | **A** | **B** | **C** | **D** | **A** | **B** | **C** | **D** | |
| 11 | 1 | −1 | −1 | 1 | 7 | 0.1 | 15 | CLDH | 72.48 |
| 12 | 1 | 1 | −1 | 1 | 7 | 0.3 | 15 | CLDH | 37.88 |
| 13 | −1 | −1 | 1 | 1 | 5 | 0.1 | 30 | CLDH | 59.80 |
| 14 | −1 | 1 | 1 | 1 | 5 | 0.3 | 30 | CLDH | 36.56 |
| 15 | 1 | −1 | 1 | 1 | 7 | 0.1 | 30 | CLDH | 47.78 |
| 16 | 1 | 1 | 1 | 1 | 7 | 0.3 | 30 | CLDH | 32.59 |

**Table 7.** Values of model coefficients for adsorptive removal of PBV.

| Main Coefficient | Value |
|---|---|
| $b_0$ | 38.24 |
| $b_1$ | −8.67 |
| $b_2$ | −5.26 |
| $b_3$ | −8.16 |
| $b_4$ | 14.24 |
| $b_{12}$ | 1.97 |
| $b_{13}$ | 4.93 |
| $b_{14}$ | −6.93 |
| $b_{23}$ | 1.77 |
| $b_{24}$ | 0.13 |
| $b_{34}$ | 0.05 |
| $b_{123}$ | −1.43 |
| $b_{124}$ | 1.18 |
| $b_{134}$ | 1.06 |
| $b_{234}$ | −0.08 |
| $b_{1234}$ | 0.04 |

### 3.3.2. Analysis of Variance (ANOVA)

ANOVA is a statistical technique that subdivides the total variation in a set of data into component parts associated with specific source of variation for adequacy and significance of predicted model. Data obtained from ANOVA analysis for the coded quadratic model at a confidence level of 95% are represented in Table 8. The table shows that the equation adequately represents the actual relationship between response and the significant variables. This is confirmed by a higher F value while the $p$ value < 0.05. In this model the F value is 361.05 and $p$ value is 0.0002 (which is less than 0.05, justifying the significance of the model). Moreover, interaction effects as significant model terms can be used for modeling the experimental system. According to the ANOVA analysis, the significant terms are the adsorbent dosage (A), solution pH (B), initial dye concentration (C), nature of LDH (D), the interaction between adsorbent dosage and pH solution (AB), the interaction between adsorbent dosage and initial dye concentration (AC) and the interaction between solution pH and initial dye concentration (BC). From the obtained fitting equation (Equation (9)), it appears that the adsorption of PBV was positively correlated to the nature of LDH, and the interaction between adsorbent dosage and solution pH, adsorbent dose and initial dye concentration, and between solution pH and initial dye concentration. Meanwhile, the increase in adsorbent dosage, solution pH and initial dye concentration resulted in a reduction of PBV adsorption.

$$Y = 38.24 - 8.67\,A - 5.26\,B - 8.16\,C + 14.24\,D + 1.97\,AB + 4.93\,AC + 1.77\,BC \tag{9}$$

**Table 8.** Analysis of variance for PBV adsorption.

| Source | Sum of Squares | df | Mean Square | F Value | *p*-Value Prob > F | |
|--------|----------------|-----|-------------|---------|--------------------|-----|
| Model | 7310.92 | 7 | 609.24 | 361.05 | 0.0002 | significant |
| A | 1203.9 | 1 | 1203.9 | 713.46 | 0.0001 | |
| B | 441.96 | 1 | 441.96 | 261.91 | 0.0005 | |
| C | 1064.86 | 1 | 1064.86 | 631.06 | 0.0001 | |
| D | 3242.99 | 1 | 3242.99 | 1921.87 | <0.0001 | |
| AB | 62.21 | 1 | 62.21 | 36.87 | 0.0090 | |
| AC | 389.18 | 1 | 389.18 | 230.64 | 0.0006 | |
| BC | 50.16 | 1 | 50.16 | 29.72 | 0.0121 | |
| Residual | 5.06 | 3 | 1.69 | | | |
| Cor Total | 7315.99 | 10 | | | | |

$R^2 = 0.980$; $R_{adj}^2 = 0.997$.

For testing significant effects of regression coefficients for the proposed model, predicted values were compared with experimental values as shown is Table 9. From the table, it can be seen that the values nearly coincide, which indicates a correspondence between the mathematical model and the experimental data. The correlation between the theoretical and experimental response calculated by the model is satisfactory. The normality of the data can be checked by plotting normal probability plot of the residuals. The data are normally distributed, when the data points on the plot fall fairly close to the straight line. Normal probability plot of residual calculated for the considered model are shown in Figure 9. The figure shows a linear relationship with high correlation coefficient which suggests good applicability of the model for the explanation of experimental data.

**Table 9.** Predicted vs. experimental response for the equilibrium adsorption capacity of PBV.

| Run | Actual | Predicted | Residual |
|-----|--------|-----------|----------|
| 1 | 20.45 | 20.76 | −0.31 |
| 2 | 14.20 | 14.82 | −0.62 |
| 3 | 72.48 | 71.59 | 0.89 |
| 4 | 59.80 | 60.11 | −0.31 |
| 5 | 48.62 | 48.00 | 0.62 |
| 6 | 92.24 | 92.86 | −0.62 |
| 7 | 32.59 | 31.97 | 0.62 |
| 8 | 15.90 | 15.59 | 0.31 |
| 9 | 47.78 | 47.74 | 0.04 |
| 10 | 40.47 | 40.50 | −0.03 |
| 11 | 32.36 | 32.32 | 0.04 |
| 12 | 37.88 | 37.57 | 0.31 |
| 13 | 11.81 | 11.85 | −0.04 |
| 14 | 36.56 | 37.45 | −0.87 |
| 15 | 22.01 | 21.11 | 0.89 |
| 16 | 26.67 | 27.56 | −0.89 |

### 3.3.3. Response Surface Analysis

Response surface methodology (RSM) was developed by considering the significant interactions in the full experimental design to optimize the critical factors and describe the nature of the response in the experiment. 3D surface plots of interaction effects are given in Figure 10. The figure presents three significant interactions between adsorbent dosage/solution pH, adsorbent dosage/dye concentration and dye concentration/solution pH. From Figure 10a, it could be seen that the equilibrium adsorption capacity of PBV increased with decreasing adsorbent dosage and solution pH. The greater adsorption efficiency was obtained at initial concentration of 15 mg/L with CLDH. However, the adsorption increased with decreasing adsorbent dosage and dye concentration (Figure 10b). The highest value was obtained at pH of 5 with CLDH. Figure 10c indicates that the PBV adsorption capacity increased when the solution pH and dye concentration decreased. The highest equilibrium adsorption capacity was obtained at adsorbent dose of 0.1 g/L with CLDH.

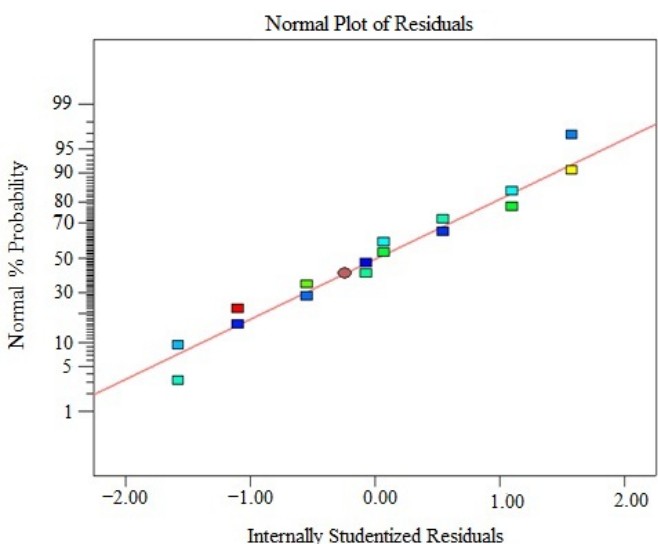

**Figure 9.** Normal probability plot of residual for the removal of PBV.

**Figure 10.** Response surface plots for PBV removal. Solution pH vs. Adsorbent dosage (**a**), dye concentration vs. Adsorbent dosage (**b**), and dye concentration vs. Solution pH (**c**).

### 3.3.4. Optimization Analysis

The optimum conditions for four variables; adsorbent dosage, solution pH, dye concentration and nature of LDH were obtained. The best conditions for the removal of PBV are obtained by calcined LDH at initial concentration of 15 mg/L, pH of 5, and adsorbent dosage of 0.1 g/L. Under these conditions, the highest equilibrium adsorption capacity was obtained. In addition, it was observed that the experimental values obtained were in good agreement with the values predicted from the models, with relatively small errors between the predicted and the experimental values, which were only 0.02%.

### 4. Conclusions

In summary, the synthesized zinc/aluminum layered double hydroxides were utilized for the removal of patent blue V dye from aqueous solution under different conditions. It can be observed that calcination of LDH strongly enhances its adsorption potential. The process is very rapid and the adsorption yield increased with an increase in the adsorbent dosage. The highest adsorption occurred in acidic medium. Kinetic data were best fitted to the pseudo-second-order kinetic model. Dye adsorption increased with the increase in the initial concentration according to Langmuir adsorption isotherm model. The adsorption process was endothermic. The effects of significant variables and their interactions in the adsorption were determined by full factorial experimental and optimum were established. The predicted values were in good agreement with the experimental with relatively small errors.

**Author Contributions:** Conceptualization, A.M., A.E. and N.B.; methodology, A.M., H.T. and Z.R.; writing—original draft, A.M. and N.T.; writing—review and editing, A.E., H.T., Z.R. and N.B.; supervision, N.B. All authors have read and agreed to the published version of the manuscript.

**Funding:** This research received no external funding.

**Institutional Review Board Statement:** Not applicable.

**Informed Consent Statement:** Not applicable.

**Data Availability Statement:** Data presented in this study are available in the article.

**Conflicts of Interest:** The authors declare no conflict of interest.

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
