# Peer review of "Patent Blue V Dye Adsorption by Fresh and Calcined Zn/Al LDH: Effect of Process Parameters and Experimental Design Optimization"

_jcs, doi:10.3390/jcs6040115_

Round 1
Reviewer 1 Report
Machrouhi et al. have studied in this paper the adsorptive removal of patent blue V dye from aqueous solution by Zn/Al layered double hydroxide in fresh and calcined forms. The adsorbent was characterized by XRD, FTIR and TGA-DTA methods. Dye removal was evaluated under different experimental conditions, which included the contact time, pH, adsorbent dosage, temperature and initial dye concentration.
Suggestion for authors:
In general: ‘maximum adsorption capacity’ has a special meaning; since it was not determined in this work, ‘maximum’ should be replaced with ‘highest’ if adsorption capacity is discussed throughout the paper.
Comments:
--- line 19-20: replace ‘maximum’ with ‘highest’;
--- line 25-26 (abstract): the statement is not true in the current form; authors can state that ‘the optimal conditions to reach high equilibrium adsorption capacity were achieved…”;
--- line 72: LDH have been prepared by several authors previously. The current paper is possibly a repetition or modification of an earlier synthetic work; therefore, authors have to provide reference to earlier works, for example, by stating that LDH was synthesized by modifying a previous procedure [ref] (if this is the case);
--- line 95: please remove ‘For’ at the beginning of the sentence;
--- Table 2: please include unit for adsorbent dosage and dye concentration;
--- lines 121-123: The sentence requires a reference to LDH XRD;
--- Figure 5, 6, 7 and 8: please provide the initial pH value;
--- line 317-328: statements here are related to the ‘equilibrium adsorption capacity’, not to the removal; please correct;
--- line 333: ‘the maximum adsorption capacity’ have to be replaced by ‘the highest equilibrium adsorption capacity’; the maximum adsorption capacity was not calculated in this work.
Author Response
Reviewer 1
Machrouhi et al. have studied in this paper the adsorptive removal of patent blue V dye from aqueous solution by Zn/Al layered double hydroxide in fresh and calcined forms. The adsorbent was characterized by XRD, FTIR and TGA-DTA methods. Dye removal was evaluated under different experimental conditions, which included the contact time, pH, adsorbent dosage, temperature and initial dye concentration.
Suggestion for authors:
In general: ‘maximum adsorption capacity’ has a special meaning; since it was not determined in this work, ‘maximum’ should be replaced with ‘highest’ if adsorption capacity is discussed throughout the paper.
Comments:
--- line 19-20: replace ‘maximum’ with ‘highest’;
Modification was done.
--- line 25-26 (abstract): the statement is not true in the current form; authors can state that ‘the optimal conditions to reach high equilibrium adsorption capacity were achieved…”;
Modifications were done line 25-26 (abstract).
--- line 72: LDH have been prepared by several authors previously. The current paper is possibly a repetition or modification of an earlier synthetic work; therefore, authors have to provide reference to earlier works, for example, by stating that LDH was synthesized by modifying a previous procedure [ref] (if this is the case);
The references were added.
--- line 95: please remove ‘For’ at the beginning of the sentence;
Modification was done.
--- Table 2: please include unit for adsorbent dosage and dye concentration;
The unit for adsorbent dosage and dye concentration were included.
--- lines 121-123: The sentence requires a reference to LDH XRD;
The modification was done.
--- Figure 5, 6, 7 and 8: please provide the initial pH value;
The initial pH value was added.
--- line 317-328: statements here are related to the ‘equilibrium adsorption capacity’, not to the removal; please correct;
The modifications were done.
--- line 333: ‘the maximum adsorption capacity’ have to be replaced by ‘the highest equilibrium adsorption capacity’; the maximum adsorption capacity was not calculated in this work.
The sentence was replaced.
Reviewer 2 Report
The study is interesting to certain audience. The experiment was well-designed. I recommend publishing this paper in the present form.
Author Response
Thank you for the review of the manuscript.
Author Response
The work has tested the effects of several parameters on dye adsorption on synthesized inorganic adsorbents. From the cited sources, self-citations make up 26% within period 2018-2021. This indicates that the authors have been dealing with analogous issues for at least 5 years and are sufficiently experienced. The authors performed a lot of experimental work including the preparation and characterization of the adsorbents and adsorption experiments with convincing results. In the adsorption process of Patent blue V dye routine parameters have been tested as effect of contact time, pH, adsorbent dose, temperature and initial dye concentration. Adsorption models and kinetics have been identified. In order to optimize the parameters, they performed a factorial analysis using statistical software. Beside the large number of statistical data produced, I lack the inclusion of an idea of ​​the mechanism of the adsorbent-adsorbate interaction, which would be a valuable complement to the study.
Here are some suggestions for improving the text:
Line 24: the number in parenthesis should be (24);
The modification was done.
Line 33, 177, 345: … This results …, … dosage increases…, ... enhances
"s" was added.
Line 52: incorrect formula ; it should be
The modification was done.
Line 73: the formulation … “with total concentration of metal” … is unclear. I recommend to specify a portion of each component more.
The metals ions were specified.
Line 93-94: I recommend to use inferior index at pH.
The modification was done.
Line 105: assign the formula numbering next to the formula in line.
The modification was done.
Line 250: ΔS0 is incorrect; ΔG0 should be.
The modification was done.
Figure 8: the legend is incomplete.
The legend in figure was completed.
Figure 9: complete a legend in the figure and correct the typing error under the figure.
The figure 9 was modified.
Author contributions are missing.
Author contributions is now added
Round 2
Reviewer 1 Report
Authors have corrected the manuscript according to reviewer's comments. I suggest to accept the manuscript in its present form.